# Time-Reversibility, Causality and Compression-Complexity

**DOI:** 10.3390/e23030327

**Published:** 2021-03-10

**Authors:** Aditi Kathpalia, Nithin Nagaraj

**Affiliations:** 1Department of Complex Systems, Institute of Computer Science of the Czech Academy of Sciences, Czech Academy of Sciences, Pod Vodárenskou věží 271/2, 182 07 Prague, Czech Republic; 2Consciousness Studies Programme, National Institute of Advanced Studies (NIAS), Indian Institute of Science Campus, Bengaluru 560012, India; nithin@nias.res.in

**Keywords:** time-reversibility, time-irreversibility, temporal asymmetry, compression-complexity, effort-to-compress, compressive potential, interventional causality, heart period variability asymmetry, sunspot numbers

## Abstract

Detection of the temporal reversibility of a given process is an interesting time series analysis scheme that enables the useful characterisation of processes and offers an insight into the underlying processes generating the time series. Reversibility detection measures have been widely employed in the study of ecological, epidemiological and physiological time series. Further, the time reversal of given data provides a promising tool for analysis of causality measures as well as studying the causal properties of processes. In this work, the recently proposed *Compression-Complexity Causality (CCC)* measure (by the authors) is shown to be free of the assumption that the "cause precedes the effect", making it a promising tool for causal analysis of reversible processes. CCC is a data-driven interventional measure of causality (second rung on the *Ladder of Causation*) that is based on *Effort-to-Compress (ETC)*, a well-established robust method to characterize the complexity of time series for analysis and classification. For the detection of the temporal reversibility of processes, we propose a novel measure called the *Compressive Potential based Asymmetry Measure*. This asymmetry measure compares the probability of the occurrence of patterns at different scales between the forward-time and time-reversed process using ETC. We test the performance of the measure on a number of simulated processes and demonstrate its effectiveness in determining the asymmetry of real-world time series of sunspot numbers, digits of the transcedental number π and heart interbeat interval variability.

## 1. Introduction

When simulated data or data recorded from real world processes are present with us, it is possible for us to create an imaginary process exhibiting the reverse dynamics of the original process. Researchers, primarily in the field of time series analysis, have widely deployed this technique to determine the time-reversibility/irreversibility of a given process. Statistical time reversibility or time symmetry implies that statistical properties of given time series remain invariant regardless of the direction of time. Statistical time irreversibility implies otherwise. Time irreversibility is a common feature of non equilibrium systems [1,2,3] as well as systems driven by non-conservative forces [4]. This has been widely observed in time series obtained from ecological, epidemiological and physiological systems. Some examples include time series recordings of measles outbreak [5,6], annual phytoplankton bloom [6], and electroencephalographic recordings of normal [7] and epileptic subjects [8,9,10] as well as beat-to-beat time interval recordings from the heart of normal and abnormal subjects [11,12,13,14].

Many methods for the detection of statistical time reversibility have been discussed in the literature. These include methods that make use of moment-based tests [15,16,17], those analyzing distance between the distributions of forward and reversed time series [8,18,19], and methods based on visibility graphs [20,21,22]. Along with inference on reversibility, some of these methods are useful in the characterization of the nature of processes, particularly to distinguish non-linearity from Gaussian noise and to provide insights into the underlying mechanisms for observed non-linear data [6,18,23,24,25,26].

In physical macroscopic systems, the arrow of time manifests as a consequence of the second law of thermodynamics, where, for an isolated system, the thermodynamic entropy of the system can only increase. This is the reason why the manifestation of events follows a particular order and not its reverse. For example, the process of a glass falling and smashing on the floor cannot be reversed. The methods discussed in the previous paragraph characterize the irreversibility of time series only in a statistical sense. Thus, without any rigorous mathematical formulation, the relation between time irreversibility and thermodynamic entropy production remained a qualitative statement for several years. In a series of papers, J.M.R. Parrondo and his group introduced Kullback Leibler divergence (KLD), as measured between the probability distributions of the forward process and its time-reversed version, as being related to the thermodynamic entropy produced by the process. More specifically, KLD, multiplied by the Boltzmann constant, is shown to be a lower bound to the entropy production along the process [27,28,29,30]. Their result is a generalization of Landauer’s principle relating entropy production to any logically irreversible manipulation of information [31].

Time-reversed processes are an important aspect of the study of causality (i.e., the determining of cause–effect relationships between given data). Causal analysis and detection of these signals can prove to be of immense use in the causal characterization of these signals, as well as helping to offer insight into the assumptions and properties of the employed causality measures. For example, the pioneering mathematical formulation for time series causality testing, Granger Causality (GC) [32], works based on the assumption that the "cause precedes the effect". Paluš et al. [33] have analyzed this assumption for GC and other causality methods, such as Conditional Mutual Information [34,35], Predictability Improvement [36] and Convergent Cross Mapping [37], by evaluating the performance of these methods on time-reversed coupled processes. While the above assumption is explicit in the formulation of GC, this is not the case for other methods. The analysis helps shed some light on the properties and hidden assumptions of these methods. It is also interesting to note the behavior of these methods for time-symmetrical (reversible) as well as asymmetric (irreversible) processes.

Recently, complexity measures have begun to be widely used for time series analysis and classification. These measures serve as great tools to offer insights into and understand time series arising from different complex systems, further helping in their prediction and control. Several measures of complexity exist, which use different approaches for complexity estimation [38]. Compression-complexity measures define the complexity of a time series by using (optimal and sometimes universal) lossless data compression algorithms. These measures are practical methods that have the advantage of efficiently characterizing the complexity of short and noisy time series over traditional complexity measures (such as Shannon Entropy [34]) [39]. Some examples of compression-complexity measures include Lempel-Ziv Complexity [40] and Effort-to-Compress (ETC) Complexity [41]. Both of these measures have been employed in a wide variety of applications, such as the analysis of neural spike trains, financial time series and cardiovascular dynamics [42,43], shaping the behaviour of feedback musical instruments [44], and for causal inference directly from pairs of genome sequences belonging to the SARS-CoV-2 virus, which could be used in epidemiological surveillance, contact-tracing, motif discovery and evolution of virulence and pathogenicity [45]. Time-series-based causality estimation techniques have also been proposed, using compression-complexity approaches such as *Compression-Complexity Causality (CCC)* [46,47] and *ORIGO* [48].

In this work, we employ selected compression-complexity approaches to analyze time-reversed processes. The motivation for this work is twofold: to test the usefulness of compression-complexity measures for applications on time reversed processes and secondly to study the properties of these compression-complexity approaches by observing their behavior in such applications. The work included in this manuscript can mainly be divided into two categories: 1. Causal analysis of time-reversed simulated processes using the CCC measure. Performance of CCC is compared with that of other widely used methods, Transfer Entropy (TE) and GC. This gives insights on the measure CCC and useful information on the applicability of the measure. 2. A novel method for detection of time irreversible processes is proposed. This method is based on the measure ETC which has several useful properties (as will be discussed in Section 2.1 below). We conjecture some relations of the proposed method to thermodynamic entropy production along a process which could be useful in the determination of the physical arrow of time. The method is first tested on simulations and then its applicability is demonstrated for real-world processes by taking three datasets: time-series of sunspot numbers, digits of the transcendental number π and heart interbeat interval recordings from young and elderly healthy human subjects.

## 2. Methods

In this section, we describe the compression-complexity measures used in this work: ETC and CCC. Finally, we propose a measure for the detection of time-reversibility of time series data which is based on ETC.

### 2.1. Effort-to-Compress

Effort-To-Compress or ETC, measures the effort needed to compress the input sequence by a lossless compression algorithm called the Non-Sequential Recursive Pair Substitution (NSRPS) algorithm [49]. The time series is first converted into a symbolic sequence. For the sake of simplicity, uniformly sized bins are taken. The NSRPS algorithm works by finding that pair of consecutive symbols in the input sequence which occurs the maximum number of times and replacing it with a new symbol in the first iteration. In next iteration, the second most frequently occurring pair is replaced with a new symbol. This procedure of replacing the most frequently occurring pair is repeated at each iteration until the length of the sequence reduces to one or the sequence turns into a constant sequence of reduced length (this will definitely happen as the length of the sequence decreases at each iteration). The ETC complexity measure is defined as “the number of iterations needed to reduce the input sequence into a constant sequence”. As an example, consider the input sequence 011101001. In the first iteration, ETC transforms 011101001→211202 (since ‘01’ is the most frequently occurring pair, it is replaced by the new symbol ‘2’). In the next iteration, ETC transforms 211202→31202 (all pairs occur equally frequently, so the first pair is replaced). Subsequently, 31202→4202→502→62→7. Thus, it took six iterations to transform 011101001 to a constant sequence (7). The value of the ETC measure, in this instance, is n=6. The normalized value of the measure is given by $\frac{n}{N-1}$, where N is the length of the input sequence (in this case, it is 0.75). $0\leq \frac{n}{N-1}\leq 1$ always. A higher value of normalized ETC implies higher complexity and a lower value implies lower complexity of time-series.

It has been shown that both Lempel Ziv and ETC complexity measures beat Shannon entropy in the accurate characterization of the dynamical complexity of both deterministic chaotic systems and stochastic (Markov) systems in the presence of noise [38,39]. Moreover, ETC is shown to have the additional advantage of reliably capturing the complexity of very short time series over the Lempel Ziv complexity measure [39].

While ETC has been utilized for the purpose of measuring the complexity of time-series using the NSRPS compression algorithm, we would like to note that there are other methods that have used NSRPS/algorithms similar to NSRPS for data compression. For example, an offline grammar-based data compression algorithm, very similar to the NSRPS scheme, called "Re-Pair", has been developed and used independently. Re-Pair has been shown to achieve high lossless compression ratios and to offer good performance for decompression, but has the drawback of high memory consumption [50].

### 2.2. Compression-Complexity Causality

CCC is a measure to estimate causality between time-series, which employs the compression-complexity measure, ETC, for its computation. It is based on an *Interventional Complexity Causality (ICC)* scheme which does not assume separability of cause and effect samples in time-series. This is unlike existing measures of time-series causality estimation, which are *associational* (a means of inferring causation, which is lower than intervention on the Ladder of Causation [51]) and computed based on directional correlations of cause–effect samples.

To estimate causality from a time series *Z* to a time series *Y*, ICC measures how the dynamics of *Z* influence the dynamics of *Y*. For this, intervention is done on the system to create new hypothetical blocks of time series data, $Z_{past}+\Delta Y$, where $Z_{past}$ is a window of length *L* samples, taken from the immediate past of the window ΔY. Here, "+" refers to appending, for, e.g., time series C=[3,4,5] and D=[a,b], C+D=[3,4,5,a,b]. ICC is defined as “the change in the dynamical complexity of time series *Y* when ΔY is seen to be generated jointly by the dynamical evolution of both $Z_{past}$ and $Y_{past}$ as opposed to by the reality of the dynamical evolution of $Y_{past}$ alone”. To estimate CCC from a time series *Z* to *Y*, we compute dynamical complexities (in ICC formulation) using the measure ETC. $CC(\Delta Y|Y_{past})$—the dynamical complexity of the current window of data, ΔY, from a time series *Y* conditioned with its own past, $Y_{past}$—is computed and compared with $CC(\Delta Y|Y_{past},Z_{past})$, the dynamical complexity of ΔY conditioned jointly with the past of both *Y* and *Z*, $\left(Y_{past},Z_{past}\right)$. Mathematically,
(1)$$CC\left(\Delta Y|Y_{past}\right)=ETC\left(Y_{past}+\Delta Y\right)-ETC\left(Y_{past}\right),$$
(2)$$CC\left(\Delta Y|Y_{past},Z_{past}\right)=ETC\left(Y_{past}+\Delta Y,Z_{past}+\Delta Y\right)-ETC\left(Y_{past},Z_{past}\right),$$
(3)$$CCC_{Z_{past}\to \Delta Y}=CC\left(\Delta Y|Y_{past}\right)-CC\left(\Delta Y|Y_{past},Z_{past}\right).$$

Averaged CCC from *Z* to *Y* over the entire length of time series, with the window ΔY being slided by a step-size of δ, is estimated as
(4)$$CCC_{Z\to Y}={\bar{CCC}}_{Z_{past}\to \Delta Y}=\bar{CC}\left(\Delta Y|Y_{past}\right)-\bar{CC}\left(\Delta Y|Y_{past},Z_{past}\right),$$
where ${\bar{CCC}}_{Z_{past}\to \Delta Y}$ denotes average CCC and $\bar{CC}\left(\cdot \right)$ denotes average CC computed from all the windows taken in the time-series.

CCC does not make assumptions such as determinism, stochasticity, stationarity, linearity, gaussianity or markovian property, and can therefore be applied widely and reliably to real-world time series data. It has also been shown to overcome the limitations of existing measures (GC, TE) in cases of signals with poor temporal resolution, long-term memory, noise, non-uniform sampling, filtering and short-length signals [46].

### 2.3. A Novel Temporal Asymmetry Measure

We first discuss the theoretical underpinnings of ETC which help us to develop a novel measure, *compressive potential*, which is the basis for temporal asymmetry measure proposed in this work.

#### 2.3.1. Theoretical Underpinnings of ETC

Though the ETC measure, as well as the underlying NSRPS algorithm (or the similar Re-pair algorithm), have been widely used, to the best of our knowledge, this theoretical understanding of the ETC measure has not been presented in any earlier work.

Let *Y* be a given sequence of symbols of length *N*, $Y=a_{1}a_{2}a_{3}a_{4}\ldots a_{N}$, where $a_{1},a_{2},a_{3},a_{4},\ldots a_{N}\in \left\{b_{1},b_{2},b_{3}\ldots b_{m}\right\}$. Thus, $a_{1},a_{2},a_{3},a_{4},\ldots a_{N}$ can be taken as one of *m* possible symbols.

We define the following for *Y*:$Y_{1}$: the given sequence *Y* as it is;$Y_{2}$: transformed *Y*, once the most frequently occurring pair in *Y* has been substituted with another symbol, i.e., the sequence after the first iteration of the ETC algorithm;$Y_{i}$: transformed sequence after i−1 iterations of the ETC algorithm, where, at each iteration, the most frequently occurring pair in that sequence is being substituted by a new symbol;$Y_{n+1}$: transformed sequence *Y* after *n* iterations, where no more iterations after this are possible, and hence *n* is the ETC value;$X_{1}$: most frequently occurring pair in $Y_{1}$. In other words, it is the first most dominant shortest *pattern* (of length 2);$X_{2}$: most frequently occurring pair in $Y_{2}$. In other words, it is the second most dominant shortest pattern (of length 2 in $Y_{2}$, but may be of length 2 or 3 in the original sequence, *Y*);$X_{i}$: most frequently occurring pair in $Y_{i}$;$X_{n}$: most frequently occurring pair in $Y_{n}$. It is the $n^{th}$ most dominant shortest pattern.
Let *W* be the event of joint occurrence of paired patterns $\left(X_{1},X_{2},X_{3},\ldots X_{n}\right)$ occurring at different levels of transformations of *Y*. The probability of joint occurrence of these events is given as
(5)$$\begin{array}{cc} p(W)= & p(X_{1},X_{2},X_{3},\ldots X_{n}),\\ = & p\left(X_{n}|X_{1},X_{2},\ldots X_{n-1}\right)\cdot p\left(X_{1},X_{2},\ldots X_{n-1}\right),\\ = & p\left(X_{n}|X_{1},X_{2},\ldots X_{n-1}\right)\cdot p\left(X_{n-1}|X_{1},X_{2},\ldots X_{n-2}\right)\cdot p\left(X_{1},X_{2},\ldots X_{n-2}\right),\\ \; & \vdots \\ = & p\left(X_{n}|X_{1},X_{2},\ldots X_{n-1}\right)\cdot p\left(X_{n-1}|X_{1},X_{2},\ldots X_{n-2}\right)\ldots p\left(X_{2}|X_{1}\right)p\left(X_{1}\right). \end{array}$$Total self-information or Shannon information (an info-theoretic quantity that helps to quantify the amount of "surprise" revealed by a variable [52]), G(W) contained in the joint occurrence of patterns $\left(X_{1},X_{2},X_{3},\ldots X_{n}\right)$ can, therefore, be written as
(6)$$\begin{array}{cc} G(W) & =-\log(p(W)),\\ \; & =-\log\left(p\left(X_{1}\right)\right)-\log\left(p\left(X_{2}|X_{1}\right)\right)-\log\left(p\left(X_{3}|X_{1},X_{2}\right)\right)\ldots -\log\left(p\left(X_{n}|X_{1},X_{2},\ldots X_{n-1}\right)\right). \end{array}$$

We approximate the conditional probability, say $p(X_{i}|X_{1},X_{2},\ldots X_{i-1})$ in the above formulation by observing the frequency of pattern, $X_{i}$, in the sequence $Y_{i}$, in which all prior replacements $X_{1},X_{2},\ldots X_{i-1}$ have been done. Thus, if $q_{1},q_{2},\ldots q_{n}$ are the frequency of occurrence of patterns $X_{1},X_{2},\ldots X_{n}$ in $Y_{1},Y_{2},\ldots Y_{n}$, respectively, then Equation (Equation 6) can be written as
(7)$$\begin{array}{cc} G(W) & =-\log\left(\frac{q_{1}}{N}\right)-\log\left(\frac{q_{2}}{N-q_{1}}\right)-\log\left(\frac{q_{3}}{N-q_{1}-q_{2}}\right)\ldots -\log\left(\frac{q_{n}}{N-q_{1}-q_{2}\ldots -q_{n-1}}\right),\\ \; & =-\log\left(\frac{q_{1}}{N}\right)\left(\frac{q_{2}}{N-q_{1}}\right)\left(\frac{q_{3}}{N-q_{1}-q_{2}}\right)\ldots \left(\frac{q_{n}}{N-q_{1}-q_{2}\ldots -q_{n-1}}\right). \end{array}$$

As seen above, ETC algorithm reduces the length of sequence at each iteration by transforming the original sequence *Y*. Thus, the *compression* achieved by the ETC algorithm at any step of the algorithm can be seen as the fractional reduction in the length of the sequence achieved at that step. Let us suppose the equivalent (or average) compression (or fractional reduction in length) being done by ETC at each iteration is denoted by *x*. Then, if the ETC algorithm takes *n* steps to stop
(8)$$\begin{array}{cc} x^{n} & =\left(\frac{q_{1}}{N}\right)\left(\frac{q_{2}}{N-q_{1}}\right)\left(\frac{q_{3}}{N-q_{1}-q_{2}}\right)\ldots \left(\frac{q_{n}}{N-q_{1}-q_{2}\ldots -q_{n-1}}\right), \end{array}$$

Taking a natural logarithm on both sides
(9)$$\begin{array}{cc} n\cdot \log(x) & =\log\left(\left(\frac{q_{1}}{N}\right)\left(\frac{q_{2}}{N-q_{1}}\right)\left(\frac{q_{3}}{N-q_{1}-q_{2}}\right)\ldots \left(\frac{q_{n}}{N-q_{1}-q_{2}\ldots -q_{n-1}}\right)\right),\\ n & =\frac{\log\left(\left(\frac{q_{1}}{N}\right)\left(\frac{q_{2}}{N-q_{1}}\right)\left(\frac{q_{3}}{N-q_{1}-q_{2}}\right)\ldots \left(\frac{q_{n}}{N-q_{1}-q_{2}\ldots -q_{n-1}}\right)\right)}{\log(x)}. \end{array}$$

Using Equation (Equation 7) in Equation (Equation 9)
(10)$$\begin{array}{cc} n & =-\frac{1}{\log(x)}\cdot G\left(W\right),\\ n & =k\cdot G(W), \end{array}$$
where k=−1/log(x). Thus, ETC can be seen as a constant multiplied by the total self-information contained in the joint occurrence of most dominant (shortest) patterns at all levels (scales) of the sequence.

From Equation (Equation 9), it can be seen that ETC for a sequence is a function of the fractional reduction in length at all steps and of the equivalent or average compression per step. The quantity $\left(\frac{q_{1}}{N}\right)\left(\frac{q_{2}}{N-q_{1}}\right)\left(\frac{q_{3}}{N-q_{1}-q_{2}}\right)\ldots$$\left(\frac{q_{n}}{N-q_{1}-q_{2}\ldots -q_{n-1}}\right)$, which is the product of fractional reductions in length, is the total compression achieved by the ETC algorithm. Let us denote this by $C_{n}$, as it is the compression achieved in *n* steps. By looking at this equation, ETC can be thought of as a dimension-like quantity, computing the effective dimension at which the patterns in a sequence appear. To clarify this, let us consider the expression for the box-counting dimension [53] for a set *s*
(11)$$\dim_{box}\left(S\right)=\lim_{\epsilon \to 0}\frac{\log(N_{d}\left(\epsilon \right))}{\log(1/\epsilon )},$$
where $N_{d}\left(\epsilon \right)$ is the number of boxes of length ϵ required to cover the set *s*. $N_{d}\left(\epsilon \right)=k_{1}\cdot {\left(\epsilon \right)}^{-\dim}$, means that $N_{d}\left(\epsilon \right)$ scales as ${\left(1/\epsilon \right)}^{\dim}$.

ETC can be thought of as a dimension with a limit on the length *N* of the sequence approaching *∞*. Let $n_{\infty }$ be the total number of ETC steps required for N→∞, and let $x_{\infty }$ denote the limit (based on the working of ETC, it is fair to assume that a limit exists) of per step compression, *x*, as N→∞.
(12)$$n_{\infty }=\lim_{N\to \infty }\frac{\log(C_{n_{\infty }})}{\log(x_{\infty })}.$$

$C_{n_{\infty }}=k_{2}\cdot x_{\infty }^{n_{\infty }}$, means that the total compression achieved by ETC for the infinite length sequence scales as $x_{\infty }^{n_{\infty }}$.

#### 2.3.2. Compressive Potential

Although ETC is useful to compare the dimension at which patterns manifest, in certain scenarios, for a given pair of sequences, we may be interested in comparing the total compression $C_{k}$ in some *k* steps, when the given sequences are transformed using the ETC (or NSRPS) algorithm. Here, we may compare the quantity $\log(C_{k})$, which is the potential to transform/compress a given sequence in the first *k* iterations of the algorithm. We name this quantity the *compressive potential*, $P_{C}$. As $C_{k}<1$, $P_{C}$ is always <0. Intuitively, for a fixed *k*, $P_{C}$ attains lower values for less compressible sequences (some examples are discussed in Section 3.2.1). Since $C_{k}$ is a function of the sequence, we denote it as $C_{k}\left(X\right)$ for the sequence *X*. Similarly $P_{C}$ is a function of *X* and also of *k*.
(13)$$\begin{array}{cc} P_{C}\left(X,k\right)= & \log(C_{k}\left(X\right)),\;\mathrm{where}\\ C_{k}\left(X\right)= & \left(\frac{q_{1}}{N}\right)\left(\frac{q_{2}}{N-q_{1}}\right)\left(\frac{q_{3}}{N-q_{1}-q_{2}}\right)\ldots \left(\frac{q_{k}}{N-q_{1}-q_{2}\ldots -q_{k-1}}\right). \end{array}$$

From Equation (Equation 9), $P_{C}\left(X,k\right)=k\cdot \log\left(x_{k}\left(X\right)\right)$, where $x_{k}$ is the equivalent per-step compression considering only the first *k* iterations of the ETC algorithm. Moreover, from Equation (Equation 7), we can obtain the relationship between total self-information contained in the patterns jointly occurring in the given sequence *X* up to the $k^{th}$ level and $P_{C}\left(X,k\right)$. Let $W_{k}$ denote the the joint occurrence of paired patterns occurring at different levels of transformation of *X*, and $G(W_{k})$ be the total self-information contained in their occurrence. Since, $G\left(W_{k}\left(X\right)\right)=-\log\left(\left(\frac{q_{1}}{N}\right)\left(\frac{q_{2}}{N-q_{1}}\right)\left(\frac{q_{3}}{N-q_{1}-q_{2}}\right)\ldots \left(\frac{q_{k}}{N-q_{1}-q_{2}\ldots -q_{k-1}}\right)\right)$ or $G\left(W_{k}\left(X\right)\right)=-\log\left(C_{k}\left(X\right)\right)$, gives us
(14)$$P_{C}\left(X,k\right)=-G\left(W_{k}\left(X\right)\right);$$

$P_{C}$ is a useful quantity to be computed for given sequences when only the patterns at higher levels (scales) are to be compared or the probabilities of shorter patterns are more relevant to our analysis. By fixing the steps of sequence transformation to *k* and computing the logarithm to the natural base, the *potential* measured is not influenced by equivalent per-step compression, which is different for each sequence. Hence, the compressive potential of the ETC algorithm for given sequences allows for a direct comparison of the frequency of occurrence of particular sections (levels) of the joint patterns in the selected steps of the algorithm.

#### 2.3.3. Compressive-Potential-Based Temporal Asymmetry Measure

The measure for temporal asymmetry, $A_{P_{C}}$, of a time series *X* is formulated using compressive potential, $P_{C}$, as follows
(15)$$A_{P_{C}}\left(X,\tau ,k\right)=P_{C}\left(X,X_{\tau },k\right)-P_{C}\left(X^{\prime },X_{\tau }^{\prime },k\right).$$

In the above equation, for binned symbolic sequence *X* of given time series of length *N*, the quantities used in the computation of $A_{P_{C}}$ are as given below
(16)$$\begin{array}{cc} X_{\tau }\left(t\right)= & X(t+\tau ),\;1\leq t\leq N-\tau ,\\ X^{\prime }\left(t\right)= & X(N-t+1),\;1\leq t\leq N-\tau ,\\ X_{\tau }^{\prime }\left(t\right)= & X^{\prime }\left(t+\tau \right),\;1\leq t\leq N-\tau . \end{array}$$

$X_{\tau }$ is the time-shifted (by τ points) to the future analog of *X*, $X^{\prime }$ is the time-reversed version of the original sequence and $X_{\tau }^{\prime }$ is the time-shifted (by τ points) to the future analog of $X^{\prime }$. For example, for a given time series 1,2,3,…,12 and τ=2, we take
(17)$$\begin{array}{cc} X= & 1,2,\ldots ,10,\\ X_{\tau }= & 3,4,\ldots ,12,\\ X^{\prime }= & 12,11,\ldots ,3,\\ X_{\tau }^{\prime }= & 10,9,\ldots 1. \end{array}$$

$P_{C}\left(X,X_{\tau },k\right)$ is the joint compressive potential of the sequences $X,X_{\tau }$ based on the total compression obtained by the ETC algorithm when it is allowed to run up to *k* iterates. $P_{C}\left(X^{\prime },X_{\tau }^{\prime },k\right)$ denotes the same for sequences $X^{\prime },X_{\tau }^{\prime }$. For the computation of joint $P_{C}$, ETC algorithm is run after encoding information contained in the given sequences into a single symbolic sequence. This is illustrated using an example. Suppose that the symbolic sequences
$$\left(\begin{array}{c} A\\ B \end{array}\right)$$
of length 4 timepoints take values
$$\left(\begin{array}{cccc} 0 & 0 & 1 & 1\\ 1 & 0 & 1 & 0 \end{array}\right),$$
after each time series block (A and B) is binned using two bins. The encoding of the new sequence is done by assigning a particular value to each column. As each row in the first column can take two values, a total of four possible combinations exist that the two rows can take together in a column. Information contained in the two rows is encoded to a single row by assigning combinations of different values in the two rows, with an encoding from ‘0’ to ‘3’. The encoding for the above sequences will be as follows
$$\left(\begin{array}{cccc} 1 & 0 & 3 & 2 \end{array}\right).$$

The value of the measure $A_{P_{c}}$ can be either positive or negative, as either forward or reversed processes can have greater compressive potential. What matters in our case is the magnitude of the difference; a larger difference implies that the statistics of the forward-time and time-reversed processes are more different, and so the the process can be classified as being time-irreversible. In order to test the significance of the obtained $A_{P_{c}}$ value for a particular kind of process, we perform surrogate analysis, as is discussed in the next section.

As discussed in the introduction, KLD between the joint probability distributions of forward-time and time-reversed process is a measure of the time-irreversibility of a given process. This is not just a statistical means of testing time-irreversibility but also has an established connection to thermodynamic entropy production along the process. $A_{P_{C}}$, computed based on joint compressive potential, also indirectly compares joint probability distributions (by taking a sequence *X* and its future $X_{\tau }$) of the given process and with its reversed version, ($X^{\prime }$ and $X_{\tau }^{\prime }$). From Equation (Equation 14), we have seen that the term $P_{C}\left(X,k\right)=-G\left(W_{k}\left(X\right)\right)$, that is, the compressive potential based on the first *k* iterations, is equal to the negative of total self-information contained in the joint occurrence of most dominant paired patterns occurring up to the first *k* levels of the transformation of the sequence. Thus, the comparison that $A_{P_{C}}$ makes is not just between simple joint probability distributions (of a time sequence and its future) but joint distributions occurring jointly at all levels (scales) of the sequence (and its future). The choice of *k* limits the point at which the scales are taken. Most often, the higher set of scales (which are the shorter patterns in the original sequence, found at a lesser number of iterations of the ETC algorithm) may be most useful for our analysis. Lower sets of scales (which are longer patterns in the original sequence, found at a greater number of iterations of the ETC algorithm), may contribute to non-requisite details and be less reliable. This is because their frequency of occurrence cannot be measured very accurately, with the possibility of their occurrence becoming limited to only a few times owing to the finite length of the sequence. For this reason, the limit *k* imposed on number of iterations is useful and helps offer reliable results.

For the above reason of the fundamental similarity of $A_{P_{C}}$ to KLD, as well as the additional beneficial features discussed, the proposed measure of temporal asymmetry is extremely promising. Though not established to date, it is expected to have relations to thermodynamic entropy production along the process.

## 3. Results

In this section, we report the results of the two performed analyses: causality testing of coupled time-reversed processes and testing of the proposed temporal asymmetry measure on simulations as well as real-world time series.

### 3.1. Causality between Coupled, Time-Reversed Processes

Causality between coupled, time-reversed processes using the measure CCC was evaluated for both stochastic as well deterministic processes. Along with CCC, we also demonstrate the behavior of the two most widely used causality measures, TE and GC. For this, coupled autoregressive processes and deterministic chaotic tent maps were simulated. Autoregressive processes of order one (AR(1)) were simulated as in [46] (Equation (Equation 15)). Let *Y* and *Z* be the dependent and independent processes, respectively,
(18)$$\begin{array}{c} Y\left(t\right)=aY\left(t-1\right)+\epsilon Z\left(t-1\right)+\varepsilon_{Y,t}\\ Z\left(t\right)=bZ\left(t-1\right)+\varepsilon_{Z,t}, \end{array}$$
where a=0.9, b=0.8, t=1 to 1000 s, sampling period = 1s. ϵ is varied from 0–0.9 in steps of 0.1. $\varepsilon_{Y},\varepsilon_{Z}=\nu \eta$ are the noise terms, where ν = noise intensity = 0.03 and η follows standard normal distribution. The performance of CCC along with that of TE and GC for the original time-series is shown in Figure 3 of [46] as mean values over 50 trials.

Linearly coupled tent maps were simulated as per the following equations. Independent process, *Z*, is generated as
(19)$$\begin{array}{c} Z(t)=2Z(t-1),\;0\leq Z(t-1)<1/2,\\ Z(t)=2-2Z(t-1),\;1/2\leq Z(t-1)\leq 1. \end{array}$$

The linearly coupled dependent process, *Y*, is as below
(20)$$\begin{array}{c} Y(t)=\epsilon Z(t)+(1-\epsilon )g(t),\\ g(t)=2Y(t-1),\;0\leq Y(t-1)<1/2,\\ g(t)=2-2Y(t-1),\;1/2\leq Y(t-1)\leq 1, \end{array}$$
where ϵ is the degree of linear coupling.

Figure 6 in [46] shows the performance of CCC and TE for the original time series from linearly coupled tent maps, as ϵ is varied.

Causality estimation from time-reversed AR and chaotic tent map processes was done as follows. Fifty trials, each of length 1000 timepoints, were taken after the elimination of 100 and 2000 transients for the above simulated AR and tent map processes, respectively. A higher number of transients for tent maps were removed because chaotic maps can have longer transient times. Let $Z^{\prime }$ denote the time-reversed independent process *Z*, and $Y^{\prime }$ the time-reversed dependent process *Y* (in case of both AR and tent systems). The equations for $Z^{\prime }$ and $Y^{\prime }$ are as given below
(21)$$\begin{array}{c} Z^{\prime }\left(t\right)=Z\left(1000-t+1\right),\;\mathrm{where}\;1\leq t\leq 1000,\\ Y^{\prime }\left(t\right)=Y\left(1000-t+1\right),\;\mathrm{where}\;1\leq t\leq 1000. \end{array}$$For time-reversed AR processes, estimated causality using CCC, TE and GC is as shown in Figure 1. For time-reversed tent map processes, estimated causality using CCC and TE is as shown in Figure 2. GC values were not estimated for tent map processes as the assumption of a linear model, for this set of processes can give spurious estimates (GC works on the assumption that the given time series come from linear autoregressive processes). Results are displayed as mean values over 50 trials. GC and TE estimation was done as in [46] for the original processes. In brief, for GC, MVGC toolbox [54] was used in its default settings and TE was estimated using the MuTE toolbox [55]. Akaike Information Criteria was used for model-order estimation with the maximum model order set to 20 in the MVGC toolbox. In the MuTE toolbox, the approach of non-uniform embedding for representation of the history of the observed processes, and the nearest-neighbor estimator for estimating the probability density functions, was used. The number of lags under consideration for observed processes was set to 5 and the maximum number of nearest neighbors to consider was set to 10. The parameters used in the computation of CCC also remain the same as used for the original processes in [46]. CCC settings used for AR processes were L=150, w=15, δ=80, B=2. CCC settings used for tent map processes were L=100, w=15, δ=80, B=8.

### 3.2. Detection of Temporal Reversibility

Before testing the performance of the proposed temporal asymmetry measure on simulations and applying it to real-world datasets, we demonstrate the behavior of the Compressive Potential, $P_{C}$, on some example cases. This is done in order to provide an intuitive understanding of what $P_{C}$ captures and how a useful temporal asymmetry measure can be built on this basis.

#### 3.2.1. Example Cases to Demonstrate the Performance of Compressive Potential

The behavior of $P_{C}$ is demonstrated for a few cases. Four symbolic sequences $X_{1}$, $X_{2}$, $X_{3}$ and $X_{4}$ are simulated as shown in Table 1. $X_{1}$ and $X_{2}$ are periodic time series, while $X_{3}$ is partly periodic, partly random, and $X_{4}$ is a fully random time series. Each of the series were simulated up to a length of 10,000. The ETC value (or the number of steps) required to compress the sequences using the ETC algorithm is also displayed in the table. For the computation of $P_{C}$ and ETC value, the time series were binned using eight uniform-sized bins.

Figure 3 shows the behavior of $P_{C}$ for each of the sequences as *k* is varied. We see that, in the case of $X_{1}$ (Figure 3a), for which the repeating patterns are of the shortest length and the sequence is fully periodic, the ETC value obtained is 3, suggesting that the patterns reappear at the third dimension. $P_{C}$ value is the highest and fastest to saturate in this case, attaining a value of −3.18 as *k* becomes equal to 3.

$X_{2}$ and $X_{3}$ have similar ETC values, 95 and 100, respectively, even though $X_{2}$ is completely periodic with period length 1000, while $X_{3}$ has a more frequently occurring, shorter periodicity of period 20, interspersed with random sequences (of length 100 timepoints). For $X_{2}$, $P_{C}$ saturates at a value of −280.65, while for $X_{3}$, $P_{C}$ saturates at a value of −317.19. If we set k=40, then at this *k*, for $X_{2}$, $P_{C}=-108.30$, while for $X_{3}$, $P_{C}=-155.17$. For the two time series, the rate at which $P_{C}$ falls is also slightly different at different values of *k*. Thus, for $X_{2}$ and $X_{3}$, even though the ETC values are not very different, $P_{C}$ values, being significantly different for k=40, indicate that there is more structure in the completely periodic $X_{2}$ at the level of shorter patterns when compared to $X_{3}$. Since $X_{4}$ is completely random, ETC value is high, equal to 4110, and the $P_{C}$ values are much lower compared to other time series. For instance, $P_{C}\left(X_{4},k=200\right)=-1186.3$ and goes on decreasing further until it is fully compressed by the ETC algorithm at k=4110. In a completely random sequence, neither shorter patterns nor longer patterns exist, making the time series highly incompressible.

#### 3.2.2. Performance of Compressive-Potential-Based Temporal Asymmetry Measure on Simulations

The following processes were simulated for the detection of temporal irreversibilty using the proposed measure $A_{P_{C}}$.

**Time-reversible** processes that were simulated include:Linear Gaussian Process (LGP), that is, Gaussian noise with distribution N(0,1);Autoregressive process of second order, AR(2)
(22)$$X\left(t\right)=0.7X\left(t-1\right)+0.2X\left(t-2\right)+0.03\epsilon_{t},$$
where *t* is the time index and $\epsilon_{t}$ is Gaussian white noise, N(0,1);Static nonlinear transformation of a first order Gaussian process, STAR(1)
(23)$$\begin{array}{cc} X(t) & =\tanh^{2}\left(Y(t)\right),\;\mathrm{where}\\ Y(t) & =0.6Y\left(t-1\right)+0.03\epsilon_{t}, \end{array}$$
where $\epsilon_{t}$ is Gaussian white noise, N(0,1).

**Time-irreversible** processes that were simulated include:Self-Exciting Threshold AR (SETAR(2;2,2)) process with two regimes, each one with second-order delays
(24)$$X\left(t\right)=\left\{\begin{array}{cc} 0.62+1.25X\left(t-1\right)-0.43X\left(t-2\right)+0.0381\epsilon_{t} & \mathrm{if}\;X(t-2)\leq 3.25\\ 2.25+1.52X\left(t-1\right)-1.24X\left(t-2\right)+0.0626\epsilon_{t} & \mathrm{otherwise}, \end{array}\right.$$
where $\epsilon_{t}$ is Gaussian white noise, N(0,1);Chaotic tent-map process
(25)$$X\left(t\right)=\left\{\begin{array}{c} 2X(t-1),\;0\leq X(t-1)<1/2,\\ 2-2X(t-1),\;1/2\leq X(t-1)\leq 1. \end{array}\right.$$

A length of 10,000 timepoints were taken for each of the above processes for the estimation of $A_{P_{C}}$. This was after discarding 1000 transients for all the processes. For the case of tent map alone, 2000 transients were discarded. For the computation of $A_{P_{C}}$ value, the given time series were symbolized using eight bins. The parameters for the measure were set as τ=500,k=500.

In order to test for the statistical significance of the obtained $A_{P_{C}}$ for each process taken, surrogate analysis was done. For this, an ensemble of surrogate data $\left\{X^{S}\right\}$, consisting of 50 realizations, was constructed from the original time series using the Iterative Amplitude Adjusted Fourier Transform (IAAFT) [56]. This method preserves the power spectrum density and amplitude distribution of the original data. Randomization of the Fourier phases results in the constructed surrogate ensemble with the null hypothesis of Gaussian linear stochastic process. Since Gaussian processes are time-reversible [23,24,57,58], our null hypothesis, $H_{0}$, is that the considered process is time-reversible. To assess the statistical significance of the $A_{P_{C}}$ of original time series, *z*-test is used to to quantify its statistical deviation from $A_{P_{C}}$ values obtained in the constructed ensemble of surrogate data. $H_{0}$ is rejected in favour of the alternate hypothesis of irreversibility, with obtained *p*-values being less than or equal to the significance level, α=0.05.

Figure 4 displays the distribution of $A_{P_{C}}$ values of surrogate data as well as a dotted line showing where the $A_{P_{C}}$ value of the original time series lies for each of the simlated processes. $A_{P_{C}}$ distribution of surrogates for all the processes was found to satisfy normality based on the Anderson–Darling test. For LGP, the null hypothesis is not rejected with *p*-value = 0.50 (Figure 4a); for AR(2) process, the null hypothesis is not rejected with *p*-value = 0.47 (Figure 4b); for STAR(1), the null hypothesis is not rejected with *p*-value = 0.13 (Figure 4c). All these processes are time-reversible processes as per the existing literature [18,19,59]. The obtained $A_{P_{C}}$ values for these processes are not found to be significant, qualifying them as reversible based on our proposed measure. For SETAR process, the null hypothesis is not rejected with *p*-value = 0.35 (Figure 4d); and for tent-map process, the null hypothesis is rejected with *p*-value = $4.6\times 10^{-8}$ (Figure 4e). Both these processes are statistically irreversible processes [19,60,61,62]. While the tent map process is classified correctly using $A_{P_{C}}$, the method fails for the simulated SETAR process.

#### 3.2.3. Performance of Compressive-Potential-Based Temporal Asymmetry Measure on Real Data

We tested the performance of the asymmetry measure on three real-world datasets. These include: 1. time series of sunspot numbers; 2. digits of π after the decimal; 3. heart period variability time series obtained from young and old healthy human subjects.

Sunspot numbersTime series analysis of sunspot numbers is an active area of research. Sunspots are regions of reduced surface temperature on the Sun’s surface that appear as spots darker than the surrounding areas. They are caused by concentrations of magnetic field flux that inhibit convection. These numbers are important in the study of the solar system as well as the activities of the sun. Monthly and annual data of sunspot activity are typically known to be nonlinear, non-Gaussian and non-stationary with characteristics of chaotic sequences [63,64,65], and, hence, these are treated as *irreversible*.The sunspot numbers used in this study were obtained from the SILSO website (www.sidc.be/silso/datafiles; accessed on 24 February 2021) as monthly mean measurements (Sunspot data from the World Data Center SILSO, Royal Observatory of Belgium, Brussels). This dataset consists of monthly data recorded starting from January, 1749. It currently comprises data from 3265 months of observation, recorded up to January, 2021. The entire dataset, comprising these 3265 datapoints, was used in our analysis. A subset of this dataset, starting from the beginning of the year 1920 to the end of year 2020, is depicted in Figure 5.Digits of the transcendental number πThe decimal expansion of number π is known to be non-repeating (transcendental irrational number). Furthermore, it is widely believed that π is a *normal* number (though proof has remained elusive till date). A number is said to be *normal* (in base *b*) if, for every positive integer *N*, all possible strings of length *N* have a frequency of occurrence $b^{-N}$. Equivalently, we can say that a normal number does not prefer a *set of patterns in its expansion over others*, and, thus, every possible pattern *occurs equally often*. A recent work by Peter Trueb [66] analyzed the first 22.4 trillion decimal digits of π and found that frequencies of sequences of lengths one, two and three are consistent with the hypothesis of π being a normal number in base-10 and base-16. We claim that this would mean that normal numbers are essentially *reversible*, since the reversed order of the digits would not change the frequency of occurrence in any way (else it would fail to be normal). We consider the decimal expansion (b=10) of π up to a 1000 digits and check its reversibility using our proposed measure.For the computation of $A_{P_{C}}$ value for 1. and 2., the time-series taken were symbolized using four bins. The number of bins taken here were fewer compared to that taken for simulated data in order to allow for more patterns to repeat (and, hence, an appropriate computation of ETC to take place) in a shorter length of available data. The parameters for the measure were set as τ=500,k=100. In order to assess the statistical significance of the obtained $A_{P_{C}}$ for each process taken, surrogate data testing was done by generating a surrogate ensemble of 50 realizations in the same manner as for simulated data (discussed in Section 3.2.2). Figure 6 displays the distribution of $A_{P_{C}}$ values of surrogate data, as well as a dotted line showing where the $A_{P_{C}}$ value of the original time series lies for real datasets 1 and 2. $A_{P_{C}}$ distribution of surrogates for both the processes was found to satisfy normality based on the Anderson–Darling test. For Sunspot numbers, the null hypothesis (of reversibility) was rejected with *p*-value = 0.04 (Figure 6a). On the other hand, for digits of π, the null hypothesis was not rejected with *p*-value = 0.13 (Figure 6b). Hence, the sunspot numbers time-series was correctly determined as being irreversible and digits of π were correctly determined as being time-reversible based on the performance of $A_{P_{C}}$.Heart period variabilityThe last set of data taken was of heart interbeat intervals from young and elderly healthy human subjects. This dataset was obtained from “Physionet: Fantasia database” [67] and was originally acquired for the study in [68]. In the study, twenty young (21–34 years old) and twenty elderly (68–85 years old) subjects underwent 120 min of continuous supine resting while watching the movie Fantasia (Disney, 1940) in order to help maintain wakefulness. During this time, continuous ECG data were recorded and digitized by sampling at 250 Hz. The heartbeats were annotated using an automated arrhythmia detection algorithm and the beat annotations were later verified by visual inspection. The occurrence of each "R" peak was noted, and the time series consisting of the time difference between successive peaks was generated. This process was repeated for each of the participants.The interbeat interval variations in the heart are characterized by an asymmetric behavior under time reversal. This is because the heart decelerates faster than it accelerates, thus resulting in heart period variability asymmetry (HPVA) [69,70]. In previous studies, the HPVA has been assessed by the use of different measures applied to the difference between two successive heart period values. These studies include [11,13,14,71]. Furthermore, it has been shown that the HPVA is influenced by pathological conditions such as heart failure [14,72] and mental disorders [73,74], as well as aging [11,75]. In particular, HPVA is found to reduce with aging [11,75].In the analysis here, we estimated the values of $A_{P_{C}}$ for each of the young and old subjects by using time series obtained by subtracting successive values of heart periods (or RR intervals) taken from the Physionet Fantasia database. The first 2500 observations of heart period were taken from each subject. The number of bins used to obtain the symbolic sequence were set to four, and the parameters for estimation of $A_{P_{C}}$ were set to τ=450,k=500. Since the $A_{P_{C}}$ can attain both positive and negative values, and a higher magnitude of the measure implies higher asymmetry, the absolute values of $A_{P_{C}}$ obtained from young and old subjects were compared. The mean and standard deviation of absolute $A_{P_{C}}$ for young subjects were found to be 27.96 and 32.99, respectively, and the mean and standard deviation of absolute $A_{P_{C}}$ for elderly subjects were found to be 11.59 and 15.82, respectively. Further, the Mann–Whitney U (single tailed) test was done to check if the magnitude of $A_{P_{C}}$ values estimated from young subjects were significantly greater than the magnitude of $A_{P_{C}}$ values estimated from elderly subjects. The null hypothesis, $H_{0}$, was that the median of the magnitude of $A_{P_{C}}$ values obtained from the young population was less than or equal to the median of the magnitude of $A_{P_{C}}$ values obtained from the old population. p<0.05 was considered statistically significant. It was found that $H_{0}$ was rejected in favour of the alternate hypothesis, with the *p* value being equal to 0.0168. This suggested that the HPV of younger subjects was more irreversible or asymmetric as compared to that of older subjects. This result is in line with the findings of existing studies.

## 4. Discussion and Conclusions

As discussed in the introduction, a causal analysis of coupled time-reversed processes helps to provide insight into the causality measures used, as well as reveal interesting properties of the coupled processes. In [33], Paluš et al. have applied Conditional Mutual Information (CMI) [34,35], (an information-theoretic approach to GC, which is equivalent to TE for a particular case [76]), on coupled time-reversed autoregressive processes *X* and *Y* generated with unidirectional coupling X→Y. With both the processes reversed in time, the measured dominant causal direction reverses and was found to be from Y→X. This result shows that, with the violation of the Granger Causality principle “cause precedes the effect”, results for CMI are altered, with the effect now seeming to be the cause. This is because GC-based methods evaluate the ability of the driver process to predict or forecast the driven process. With the driver and the driven now interchanged, the information about the driver first occurs in the driven process.

A unidirectionally coupled set of Rössler systems was also evaluated in [33] using a time-reversed time series of the coupled variable. Causality was estimated using measures Convergent Cross Mapping (CCM) [37] (which is based on the topology of dynamical systems), Predictability Improvement (PI) [36] (which is a generalization of the GC principle for nonlinear dynamical systems) and CMI. It was found, using the three measures, that the discovered causal direction remained the same as for the original processes. It is suggested that this result was expected for the measure CCM, as it determines the ability of the driven system to provide information regarding the present state of the driver. Therefore, the sequence of cause and effect do not make a difference for CCM. Other measures (CMI and PI) were not expected to perform symmetrically, and hence a speculated rationale for this result is the presence of dynamical memory in chaotic systems.

CCM and PI cannot be applied to AR processes, as their working is based on the manifold (or geometry) of dynamical systems. At the same time, GC is a failure when applied to dynamical systems, as it makes the assumption of linear AR processes underlying the system. CCC is, however, a method that works for both stochastic as well as deterministic, linear and non-linear processes (this has been demonstrated in [46]). Analysis of time-reversed coupled processes in this work was thus done in order to analyse the properties of the promising measure CCC.

In consensus with the results reported in [33] for CMI, it is observed from results in Section 3.1, that, for TE and GC, the dominant causal direction for the time-reversed AR processes is reversed (compared to the original case), with the estimated values from $Z^{\prime }\to Y^{\prime }$ being less than that for $Y^{\prime }\to Z^{\prime }$. Interestingly, for the measure CCC, the trend of causality values remains unaltered when compared to the original case. To the best of our knowledge, CCC is the only measure which performs in this way for stochastic linear AR processes. Though based on Wiener’s principle, it works by measuring the change in the dynamical compression-complexity of the driven process when information from the driving process is brought to the former and is not based on prediction (in a sequential sense) of the future of the caused based on the past of the causal (as can be seen from its formulation given in Section 2.2 and discussed in detail in [46]).

For time-reversed tent map processes, for both TE and CCC, the dominant direction of causality is as identified for the original case. TE displays some spurious results at ϵ=0.4,0.5, but the trend in the magnitude of CCC values clearly increases for increasing coupling until the processes are synchronized. The values obtained for CCC are negative, which is in line with the CCC values for original coupled time series (please see [46]). The unchanging nature of both CCC and TE on the reversal of these chaotic processes may be the result of memory in dynamical systems, as has been suggested in [33].

By taking the simulated experiments results discussed above into account, it is found that the CCC measure works symmetrically for time-forward and time-reversed processes. This behavior is observed for both stochastic AR as well as deterministic chaotic processes, unlike other measures, such as TE (or CMI), where symmetric behavior is noticed only for chaotic processes with dynamical memory. These results indicate that the violation of the GC assumption that the "cause should precede the effect" does not affect the CCC measure. Thus, CCC can be applied to a broader range of processes such as microscopic processes, which are reversible, and the arrow of time is not restricted to a single direction. This also allows for the possibility of CCC being applied on quantum processes or subjective psychological processes. While quantum mechanics allows for certain processes and information to travel backwards in time, subjective experiences such as dreams and intuitions seem to dismiss the sequential order of a linear arrow of time. The possibility of these applications are, as of now, very speculative and would require appropriate translation of the data for implementation of CCC.

A reason as to why CCC works symmetrically is because the measure of ETC that it employs works on finding the most frequently occurring pairs in the given sequence in several iterations, transforming the original sequence at each iteration. Whether we run the ETC algorithm on forward-time or reverse-time windows of data (for CCC computation) does not typically make much of a difference, as the chosen pairs for substitution remain more or less the same. In measuring the causality from Z→Y, what is important is the choice of the length of past windows $Z_{past},Y_{past}$ and the future window ΔY, to which there is a potential effect of the latter two (see Section 2.2 for details). As long as these windows are selected appropriately using the parameter selection criteria for CCC (see Supplementary Material of [46] for parameter selection criteria and its details), computation of ETC and, hence, dynamical compression-complexity, which are based on the occurrence of patterns held together in these windows and not on whether the patterns occur on parsing from right to left or left to right in a sequence, determine the requisite causal influence. What this means is that once an intervention has been made at the correct spot to put appropriate blocks of cause and effect dynamics from a given time series together, whether the complexity of blocks is measured in forward-time or in reverse-time does not matter.

In the second part of this work, we proposed a compressive-potential-based temporal asymmetry measure for time series data, which is based on the ETC algorithm. It helps to compare the forward- and reverse-time joint probability distributions occurring jointly at different scales (levels) of the given time series. The established relation of the proposed compressive potential measure to the total self-information contained in the joint occurrence of most dominant paired patterns brings the asymmetry measure closer to KLD measure of temporal asymmetry. KLD is not just a statistical asymmetry measure, but is also shown to be related to thermodynamic entropy production along a process, relating the obtained value of asymmetry to the physical arrow of time for the process. KLD already has some estimators based on compression algorithms, such as the Ziv-Merhav estimator [30,77]. The proposed measure is promising because of the advantages of ETC, such as its better performance on short and noisy time series when compared to other complexity estimators [39,41]. Additionally, some of the discussed theoretical properties of ETC, by which it can account for distributions at many scales of the time series, can help to provide reversibility/irreversibility information that may be hidden at different scales. The choice of the parameter *k* in $P_{C}$ can help to fix the number of most dominant scales to take at which the most dominant probabilities are considered. The measure can also be generalized to consider some set of intermediate steps of the ETC algorithm instead of the first *k* steps, in order to compute compressive potential based on probabilities only at these intermediate specific scales.

While the proposed measure is similar to the KLD measure of temporal asymmetry, it evidently has properties different from data-driven predictability-based asymmetry estimation schemes such as the one discussed in [14]. The latter scheme exploits local nonlinear prediction properties of the time series in the time-forward and time-reverse directions. If there is a mismatch in the level of forward prediction and backward prediction of samples in the time series based on the formulated method, then the time-series is considered to be irreversible. The idea behind $A_{P_{C}}$, on the other hand, is that of mismatch in joint *compressibility* (or effort to compress) of a time-series window and a window from its future, in the forward-time and time-reverse directions. Other data-driven markers of irreversibility, such as the one discussed and applied in [75,78,79,80], have looked at the gain/slope of the relationship between two variables. More specifically, in the above studies, to assess a physiological variable that is termed “Baroreflex Asymmetry” [81,82,83], the measure looks at the relationship between heart period (HP) and systolic arterial pressure (SAP) observations. The mean slope (or ramp) of contemporaneous HP and SAP increases was compared with that of the mean slope of contemporaneous HP and SAP, which decreases. $A_{P_{C}}$, in its current stage, can be applied only to assess the asymmetry of time series from a single variable. This is the way most existing methods of asymmetry work. In the future, we would like to work towards extending the functionality of the measure to capture the asymmetry resulting from the relationship between two or more variables.

Though the use of ETC ensures that CCC works symmetrically for time-forward and time-reversed time series, still, with appropriate modification of ETC, we can obtain the temporal asymmetry measure, $A_{P_{C}}$. This measure uses $P_{C}$. Though derived from the ETC algorithm, $P_{C}$ ensures that minute differences in jointly occurring paired patterns are captured. $P_{C}\left(X,k\right)=\log\left(C_{k}\left(X\right)\right)$, thereby taking into account the actual compression taking place at each step of the ETC algorithm. In fact, the number of steps of the ETC algorithm are held constant, and the amount by which the given sequence is compressible in these steps is computed. This provides a fair ground for comparison of sequences as differences in their equivalent per-step compression, that is, the quantity *x* in Equation (Equation 9), may eventually result in their similar ETC values (this is the case for sequences $X_{2}$ and $X_{3}$, discussed as examples in Section 3.2.1). ETC is not concerned about the “scales” at which the compression of the sequence takes place; it is only concerned about the "effective dimension" at whicch the compression is taking place. Restraining the number of steps to take, through *k* (which is actually deciding which scales to consider), helps to enhance the difference or asymmetry, thus resulting in a measure adequate for capturing time-reversibility for given data. To explain this in simpler terms, the route to compressibility using ETC or NSRPS may be different for different sequences, even though the final ETC value is the same. Thus, for a time-series to be truly reversible, these routes also should be the same. The $P_{C}$ measure helps to account for this route to compressibility.

Out of the processes simulated for the testing of $A_{P_{C}}$, correct reversibility/irreversibility is detected for all the cases except for SETAR, which was incorrectly classified as being reversible. Time-irreversibility was also correctly identified for real-world time series of sunspot numbers and reversibility, correctly identified for digits of the transcendental number π. Furthermore, in line with the existing literature, the HPV of younger human subjects was found to possess greater asymmetry when compared with that of older subjects. In future work, different values of parameters *k* and τ will be taken to check for improved results using $A_{P_{C}}$. Other processes, such as continuous-time chaotic processes (Lorenz, Rössler, etc.), will be tested for reversibility. A generalized version of the measure using intermediate range of ETC steps for the computation of $P_{C}$ will also be tested on simulations for irreversibility detection at different scales. Further, the measure will also be tested on other real-world datasets such as ecological and epidemiological time series. The difference between $P_{c}$ values as a means to compare distributions (not just forward-time and reverse-time distributions of a single time series) is also left for future work. It would also be interesting to explore the relationship between $P_{c}$ and different types of fractal dimensions.

## Figures and Tables

**Figure 1 entropy-23-00327-f001:**
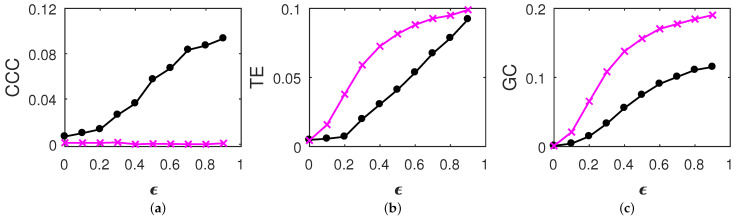
Mean causality values estimated using (**a**) CCC, (**b**) TE and (**c**) GC for coupled time-reversed AR(1) processes, from $Z^{\prime }$ to $Y^{\prime }$ (solid-line circles, black) and $Y^{\prime }$ to $Z^{\prime }$ (solid-line crosses, magenta) as the degree of coupling, ϵ is varied. CCC is invariant to time reversal, while for TE and GC, the dominant direction of causality is seen to be reversed.

**Figure 2 entropy-23-00327-f002:**
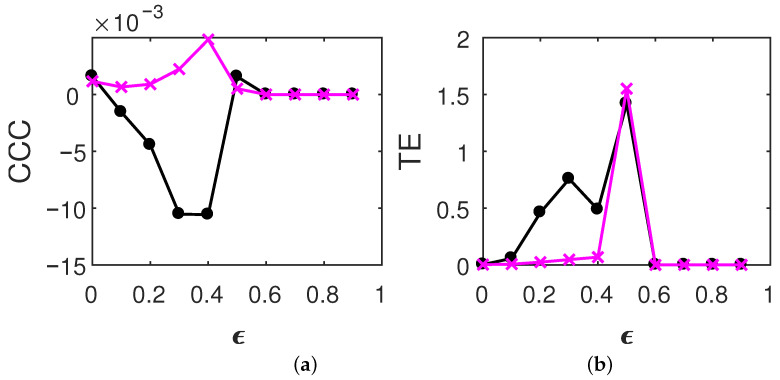
Mean causality values estimated using (**a**) CCC and (**b**) TE for linearly coupled time-reversed tent map processes, from $Z^{\prime }$ to $Y^{\prime }$ (solid line-circles, black) and $Y^{\prime }$ to $Z^{\prime }$ (solid line-crosses, magenta) as the degree of coupling, ϵ is varied. CCC is invariant to time reversal and, in the case of TE, the dominant direction of causality identified is same as for the original processes.

**Figure 3 entropy-23-00327-f003:**
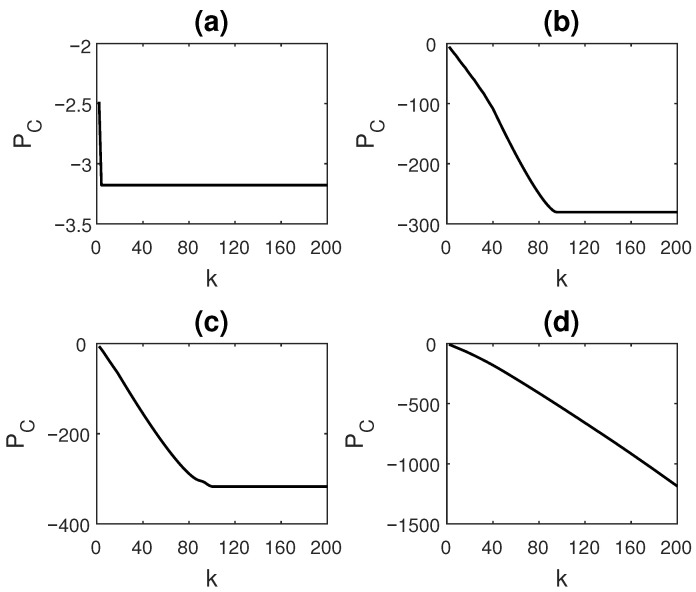
Variation in Compressive potential, $P_{C}$, with *k* for time series (**a**) $X_{1}$ (periodic with short patterns), (**b**) $X_{2}$ (periodic with long patterns), (**c**) $X_{3}$ (partly periodic, partly random) and (**d**) $X_{4}$ (completely random), simulated as per Table 1.

**Figure 4 entropy-23-00327-f004:**
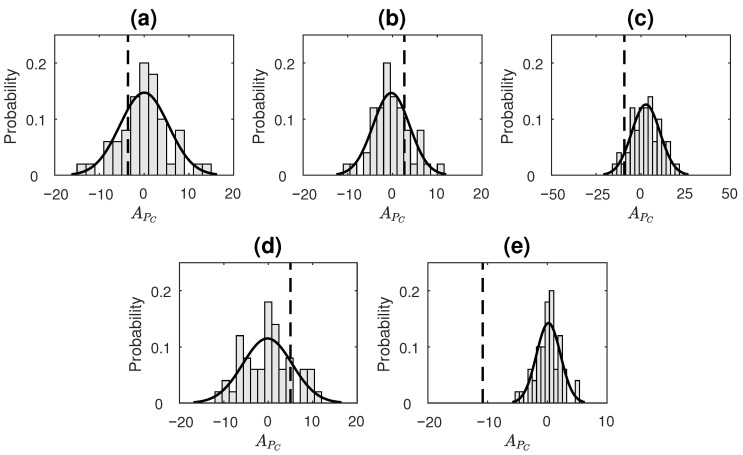
Compressive-potential-based temporal asymmetry test result on simulated data from processes: (**a**) LGP, (**b**) AR(2), (**c**) STAR(1), (**d**) SETAR(2;2,2) (**e**) Tent map. Dashed line indicates $A_{P_{c}}$ value obtained for original series. Its position is indicated with respect to Gaussian=curve-fitted normalized histogram of surrogate $A_{P_{C}}$ values that form the null hypothesis of reversible processes. Null hypothesis is not rejected in case of (**a**)–(**d**) and rejected in case of (**e**).

**Figure 5 entropy-23-00327-f005:**
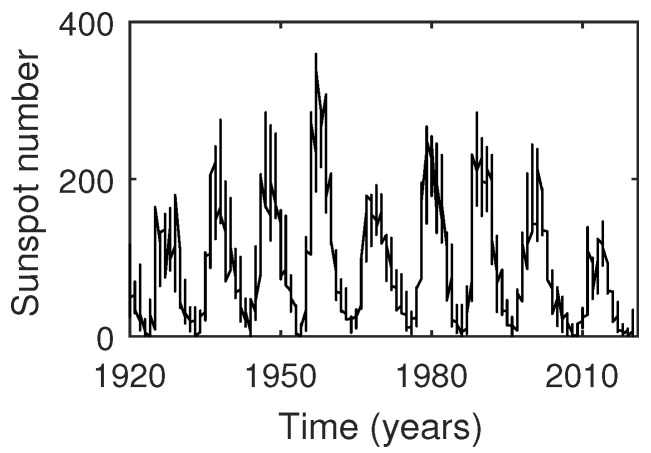
Variation in Sunspot numbers with time. Monthly mean total sunspot numbers are plotted from the beginning of the year 1920 to the end of the year 2020. Data source: www.sidc.be/silso/datafiles; accessed on 24 February 2021.

**Figure 6 entropy-23-00327-f006:**
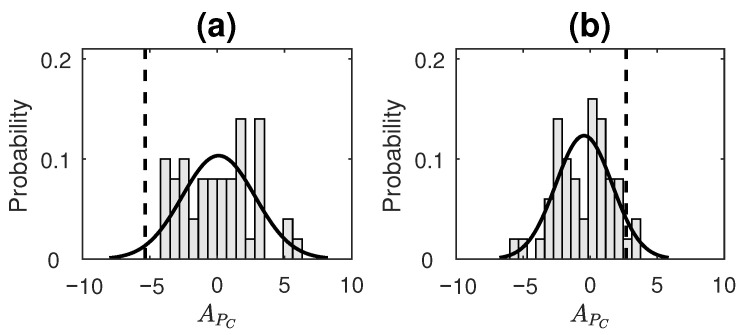
Compressive-potential-based temporal asymmetry test result on time series of (**a**) Sunspot numbers, and (**b**) Digits of π. Dashed line indicates $A_{P_{c}}$ value obtained for original series. Its position is indicated with respect to Gaussian-curve-fitted normalized histogram of surrogate $A_{P_{C}}$ values that form the null hypothesis of reversible processes. The null hypothesis is rejected in the case of (**a**) and not rejected in the case of (**b**).

**Table 1 entropy-23-00327-t001:** Time series simulated to study properties of $P_{C}$.

Time Series	Composed of	ETC
$X_{1}$	Repeating periodic sequence: [1234]	3
$X_{2}$	Repeating periodic sequence: [123…1000]	95
$X_{3}$	Repeating partly periodic partly random sequence: [123…20]
	followed by 100 random numbers uniformly chosen from between 1 and 20	100
$X_{4}$	Uniformly randomly distributed real numbers in the range (0,1)	4110

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
