# Peer review of "Time-Reversibility, Causality and Compression-Complexity"

_entropy, 2021, doi:10.3390/e23030327_

Round 1
Reviewer 1 Report
The recently proposed compression-complexity causality (CCC) method was adapted to detect time reversibility/irreversibility and tested over simulations.
The study is interesting but its originality and practical value should be better emphasized and highlighted.
- The data-driven predictability-based scheme for testing time irreversibility was exploited in ref #14. This scheme deserves to be highlighted in discussion to better understand the novelty of the present approach compared a data-driven predictability scheme.
- The null hypothesis of time irreversibility should be formally tested against the null hypothesis of time reversibility. Some suggestions are present in ref #14. A formal test is needed to better understand the possibility given by the proposed method in real contexts. Examples given in the present study is heavily irreversible, while irreversibility is much weaker in real e.g. biological and physiological processes.
- Data-driven simple markers of irreversibility have been utilized in B. De Maria et al, Am J Physiol, 317, R539-R551, 2019 in a physiological context. An interesting hypothesis was suggested: the asymmetric temporal properties are the consequence of asymmetric gain features. This issue deserves some comments in discussion given that the approach described in this study might be less efficient over asymmetries based on gain of the relation.
- Practical applications should be suggested and representative examples taken from the real world must be provided to better understand the potentiality of the method.
Reviewer 2 Report
n this manuscript the authors continue their previous work on time reversibility and causality by considering two measures - CCC measure (Compression-Complexity Causality measure) and CPA measure ( Compressive Potential based Asymmetry measure). The authors use the standard way to show that the measures work (the standard way is based on artificially generated time series and on comparison of results of application of the measure to the results of application of other measures). The measures can be of interest for the readers and because of this my opinion about the publication of the manuscript is positive.
Round 2
Reviewer 1 Report
The manuscript was improved. The authors replied satisfactorily to all my issues and followed carefully the suggestions given.